# DIRICHLET VARIATIONAL AUTOENCODER

## ABSTRACT

This paper proposes Dirichlet Variational Autoencoder (DirVAE) using a Dirichlet prior for a continuous latent variable that exhibits the characteristic of the categorical probabilities. To infer the parameters of DirVAE, we utilize the stochastic gradient method by approximating the Gamma distribution, which is a component of the Dirichlet distribution, with the inverse Gamma CDF approximation. Additionally, we reshape the component collapsing issue by investigating two problem sources, which are decoder weight collapsing and latent value collapsing, and we show that DirVAE has no component collapsing; while Gaussian VAE exhibits the decoder weight collapsing and Stick-Breaking VAE shows the latent value collapsing. The experimental results show that 1) DirVAE models the latent representation result with the best log-likelihood compared to the baselines; and 2) DirVAE produces more interpretable latent values with no collapsing issues which the baseline models suffer from. Also, we show that the learned latent representation from the DirVAE achieves the best classification accuracy in the semi-supervised and the supervised classification tasks on MNIST, OMNIGLOT, and SVHN compared to the baseline VAEs. Finally, we demonstrated that the DirVAE augmented topic models show better performances in most cases.

## 1 INTRODUCTION

A *Variational Autoencoder* (VAE) (Kingma & Welling, 2014c) brought success in deep generative models (DGMs) with a Gaussian distribution as a prior distribution (Jiang et al., 2017; Miao et al., 2016; 2017; Srivastava & Sutton, 2017). If we focus on the VAE, the VAE assumes the prior distribution to be $\mathcal{N}(\mathbf{0}, \boldsymbol{I})$ with the learning on the approximated $\hat{\boldsymbol{\mu}}$ and $\hat{\boldsymbol{\Sigma}}$. Also, *Stick-Breaking VAE* (SBVAE) (Nalisnick & Smyth, 2017) is a nonparametric version of the VAE, which modeled the latent dimension to be infinite using a stick-breaking process (Ishwaran & James, 2001).

While these VAEs assume that the prior distribution of the latent variables to be continuous random variables, recent studies introduce the approximations on discrete priors with continuous random variables (Jang et al., 2017; Maddison et al., 2017; Rolfe, 2017). The key of these approximations is enabling the backpropagation with the reparametrization technique, or the stochastic gradient variational Bayes (SGVB) estimator, while the modeled prior follows a discrete distribution. The applications of these approximations on discrete priors include the prior modeling of a multinomial distribution which is frequently used in the probabilistic graphical models (PGMs). Inherently, the multinomial distributions can take a Dirichlet distribution as a conjugate prior, and the demands on such prior have motivated the works like Jang et al. (2017); Maddison et al. (2017); Rolfe (2017) that support the multinomial distribution posterior without explicit modeling on a Dirichlet prior.

When we survey the work with a explicit modeling on the Dirichlet prior, we found a frequent approach such as utilizing a softmax Laplace approximation (Srivastava & Sutton, 2017). We argue that this approach has a limitation from the multi-modality perspective. The Dirichlet distribution can exhibit a multi-modal distribution with parameter settings, see Figure 1, which is infeasible to generate with the Gaussian distribution with a softmax function. Therefore, the previous continuous domain VAEs cannot be a perfect substitute for the direct approximation on the Dirichlet distribution.

Utilizing a Dirichlet distribution as a conjugate prior to a multinomial distribution has an advantage compared to the usage of a softmax function on a Gaussian distribution. For instance, Figure 1 illustrates the potential difficulties in utilizing the softmax function with the Gaussian distribution.

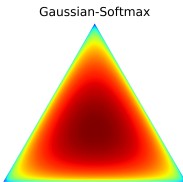 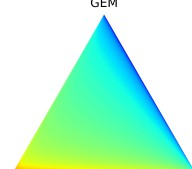 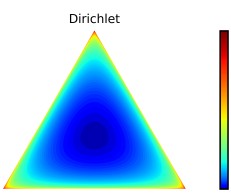

Figure 1: Illustrated probability simplex with Gaussian-Softmax, GEM, and Dirichlet distributions. Unlike the Gaussian-Softmax or the GEM distribution, the Dirichlet distribution is able to capture the multi-modality that illustrates multiple peaks at the vertices of the probability simplex.

Given the three-dimensional probability simplex, the Gaussian-Softmax distribution cannot generate the illustrated case of the Dirichlet distribution with a high probability measure at the vertices of the simplex, i.e. the *multi-modality* where the necessity was emphasized in Hoffman & Johnson (2016). Additionally, the Griffiths-Engen-McCloskey (GEM) distribution (Pitman, 2002), which is the prior distribution of the SBVAE, is difficult to model the multi-modality because the sampling procedure of the GEM distribution is affected by the *rich-get-richer* phenomenon, so a few components tend to dominate the weight of the samples. This is different from the Dirichlet distribution that does not exhibit such phenomenon, and the Dirichlet distribution can fairly distribute the weights to the components, and the Dirichlet distribution is more likely to capture the multi-modality by controlling the prior hyper-parameter (Blei et al., 2003). Then, we conjecture that an enhanced modeling on Dirichlet prior is still needed 1) because there are cases that the Gaussian-Softmax approaches, or the softmax Laplace approximation, cannot imitate the Dirichlet distribution; and 2) because the nonparametric approaches could be influenced by the biases that the Dirichlet distribution does not suffer from.

Given these motivations for modeling the Dirichlet distribution with the SGVB estimator, this paper introduces the *Dirichlet Variational Autoencoder* (DirVAE) that shows the same characteristics of the Dirichlet distribution. The DirVAE is able to model the multi-modal distribution that was not possible with the Gaussian-Softmax and the GEM approaches. These characteristics allow the DirVAE to be the prior of the discrete latent distribution, as the original Dirichlet distribution is.

Introducing the DirVAE requires the configuration of the SGVB estimator on the Dirichlet distribution. Specifically, the Dirichlet distribution is a composition of the Gamma random variables, so we approximate the inverse Gamma cumulative distribution function (CDF) with the asymptotic approximation. This approximation on the inverse Gamma CDF becomes the component of approximating the Dirichlet distribution. We compared this approach to the previously suggested approximations, i.e. approaches with the Weibull distribution and with the softmax Gaussian distribution, and our approximation shows the best log-likelihood among the compared approximations.

Moreover, we report that we had to investigate the *component collapsing* along with the research on DirVAE. It has been known that the *component collapsing* issue is resolved by the SBVAE because of the meaningful decoder weights from the latent layer to the next layer. However, we found that SBVAE has *latent value collapsing* issue resulting in many near-zero values on the latent dimensions that leads to the incomplete utilization of the latent dimension. Hence, we argue that Gaussian VAE (GVAE) suffers from the *decoder weight collapsing*, previously limitedly defined as *component collapsing*; and SBVAE has a problem of the *latent value collapsing*. Finally, we suggest that the definition of *component collapsing* should be expanded to represent both cases of *decoder weight* and *latent value collapsings*. The proposed DirVAE shows neither the near-zero decoder weights nor the near-zero latent values, so the reconstruction uses the full latent dimension information in most cases. We investigated this issue because our performance gain comes from resolving the expanded version of the *component collapsing*. Due to the component collapsing issues, the existing VAEs have less meaningful latent values or could not effectively use its latent representation. Meanwhile, DirVAE does not have component collapsing due to the multi-modal prior which possibly leads to superior qualitative and quantitative performances. We experimentally showed that the DirVAE has more meaningful or disentangled latent representation by image generation and latent value visualizations.

Technically, the new approximation provides the closed-form loss function derived from the evidence lower bound (ELBO) of the DirVAE. The optimization on the ELBO enables the representation learning with the DirVAE, and we test the learned representation from the DirVAE in two folds. Firstly, we test the representation learning quality by performing the supervised and the semi-supervised classification tasks on MNIST, OMNIGLOT, and SVHN. These classification tasks conclude that DirVAE has the best classification performances with its learned representation. Secondly, we test the applicability of DirVAE to the existing models, such as topic models with DirVAE priors on 20Newsgroup and RCV1-v2. This experiment shows that the augmentation of DirVAE to the existing neural variational topic models improves the perplexity and the topic coherence, and most of best performers were DirVAE augmented.

## 2 PRELIMINARIES

### 2.1 VARIATIONAL AUTOENCODERS

A VAE is composed of two parts: a generative sub-model and an inference sub-model. In the generative part, a probabilistic decoder reproduces $\hat{\mathbf{x}}$ close to an observation $\mathbf{x}$ from a latent variable $\mathbf{z} \sim p(\mathbf{z})$, i.e. $\mathbf{x} \sim p_\theta(\mathbf{x}|\mathbf{z}) = p_\theta(\mathbf{x}|\boldsymbol{\zeta})$ where $\boldsymbol{\zeta} = \text{MLP}(\mathbf{z})$ is obtained from a latent variable $\mathbf{z}$ by a multilayer perceptron (MLP). In the inference part, a probabilistic encoder outputs a latent variable $\mathbf{z} \sim q_\phi(\mathbf{z}|\mathbf{x}) = q_\phi(\mathbf{z}|\boldsymbol{\eta})$ where $\boldsymbol{\eta} = \text{MLP}(\mathbf{x})$ is computed from the observation $\mathbf{x}$ by a MLP. Model parameters, $\theta$ and $\phi$, are jointly learned by optimizing the below ELBO with the stochastic gradient method through the backpropagations as the ordinary neural networks by using the SGVB estimators on the random nodes.

$$\log p(\mathbf{x}) \geq \mathcal{L}(\mathbf{x}) = \mathbb{E}_{q_\phi(\mathbf{z}|\mathbf{x})}[\log p_\theta(\mathbf{x}|\mathbf{z})] - \text{KL}(q_\phi(\mathbf{z}|\mathbf{x})||p_\theta(\mathbf{z})) \tag{1}$$

In GVAE (Kingma & Welling, 2014c), the prior distribution of $p(\mathbf{z})$ is assumed to be a standard Gaussian distribution. In SBVAE (Nalisnick & Smyth, 2017), the prior distribution becomes a GEM distribution that produces samples with a Beta distribution and a stick-breaking algorithm.

### 2.2 DIRICHLET DISTRIBUTION AS A COMPOSITION OF GAMMA RANDOM VARIABLES

The Dirichlet distribution is a composition of multiple Gamma random variables. Note that the probability density functions (PDFs) of Dirichlet and Gamma distributions are as follows:

$$\text{Dirichlet}(\mathbf{x};\boldsymbol{\alpha}) = \frac{\Gamma(\sum \alpha_k)}{\prod \Gamma(\alpha_k)} \prod x_k^{\alpha_k-1}, \ \text{Gamma}(x;\alpha,\beta) = \frac{\beta^\alpha}{\Gamma(\alpha)} x^{\alpha-1} e^{-\beta x} \tag{2}$$

where $\alpha_k, \alpha, \beta > 0$. In detail, if there are $K$ independent random variables following the Gamma distributions $X_k \sim \text{Gamma}(\alpha_k, \beta)$ or $\mathbf{X} \sim \text{MultiGamma}(\boldsymbol{\alpha}, \beta \cdot \mathbf{1}_K)$ where $\alpha_k, \beta > 0$ for $k = 1, \cdots, K$, then we have $\mathbf{Y} \sim \text{Dirichlet}(\boldsymbol{\alpha})$ where $Y_k = X_k / \sum X_i$. It should be noted that the rate parameter, $\beta$, should be the same for every Gamma distribution in the composition. Then, the KL divergence can be derived as the following:

$$\text{KL}(Q||P) = \sum \log \Gamma(\alpha_k) - \sum \log \Gamma(\hat{\alpha}_k) + \sum (\hat{\alpha}_k - \alpha_k)\psi(\hat{\alpha}_k) \tag{3}$$

for $P = \text{MultiGamma}(\boldsymbol{\alpha}, \beta \cdot \mathbf{1}_K)$ and $Q = \text{MultiGamma}(\hat{\boldsymbol{\alpha}}, \beta \cdot \mathbf{1}_K)$ where $\psi$ is a digamma function. The detailed derivation is provided in Appendix B.

### 2.3 SGVB FOR GAMMA RANDOM VARIABLE AND APPROXIMATION ON DIRICHLET DISTRIBUTION

This section discusses several ways of approximating the Dirichlet random variable; or the SGVB estimators for the Gamma random variables which compose a Dirichlet distribution. Utilizing SGVB requires a differentiable non-centered parametrization (DNCP) for the distribution (Kingma & Welling, 2014d). The main SGVB for Gamma random variables, used in DirVAE, is using the inverse Gamma CDF approximation explained in the next section. Prior works include two approaches: the use of the Weibull distribution and the softmax Gaussian distribution, and the two approaches are explained in this section.

**Approximation with Weibull distribution.** Because of the similar PDFs between the Weibull distribution and the Gamma distribution, some prior works used the Weibull distribution as a posterior distribution of the prior Gamma distribution (Zhang et al., 2018):

$$\text{Weibull}(x; k, \lambda) = \frac{k}{\lambda} \left(\frac{x}{\lambda}\right)^{k-1} e^{-(x/\lambda)^k} \text{ where } k, \lambda > 0 . \tag{4}$$

The paper Zhang et al. (2018) pointed out that there are two useful characteristics when approximating the Gamma distribution with the Weibull distribution. One useful property is that the KL divergence expressed in a closed form, and the other is the simple reparametrization trick with a closed form of the inverse CDF from the Weibull distribution. However, we noticed that the Weibull distribution has a component of $e^{-(x/\lambda)^k}$, and the Gamma distribution does not have the additional power term of $k$ in the component. Since $k$ is placed in the exponential component, small changes on $k$ can cause a significant difference that limits the optimization.

**Approximation with softmax Gaussian distribution.** As in MacKay (1998); Srivastava & Sutton (2017), a Dirichlet distribution can be approximated by a softmax Gaussian distribution by using a softmax Laplace approximation. The relation between the Dirichlet parameter $\boldsymbol{\alpha}$ and the Gaussian parameters $\boldsymbol{\mu}, \boldsymbol{\Sigma}$ is explained as the following:

$$\mu_k = \log \alpha_k - \frac{1}{K} \sum_i \log \alpha_i, \ \Sigma_k = \frac{1}{\alpha_k} \left(1 - \frac{2}{K}\right) + \frac{1}{K^2} \sum_i \frac{1}{\alpha_i} , \tag{5}$$

where $\boldsymbol{\Sigma}$ is assumed to be a diagonal matrix, and we use the reparametrization trick in the usual GVAE for the SGVB estimator.

## 3 MODEL DESCRIPTION

Along with the inverse Gamma CDF approximation, we describe two sub-models in this section: the generative sub-model and the inference sub-model. Figure 2 describes the graphical notations of various VAEs and the neural network view of our model.

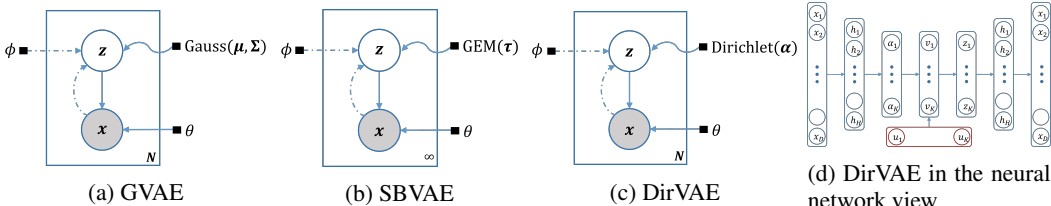

|     |     |     |     |
| --- | --- | --- | --- |
| (a) GVAE | (b) SBVAE | (c) DirVAE | (d) DirVAE in the neural network view |

Figure 2: Sub-figures 2a, 2b, and 2c are the graphical notations of the VAEs as latent variable models. The solid lines indicate the generative sub-models where the waved lines denote a prior distribution of the latent variables. The dotted lines indicate the inference sub-models. Sub-figure 2d denotes a neural network structure corresponding to Sub-figure 2c. Red nodes denote the random nodes which allow the backpropagation flows to the input.

**Generative sub-model.** The key difference between the generative models between the DirVAE and the GVAE is the prior distribution assumption on the latent variable **z**. Instead of using the standard Gaussian distribution, we use the Dirichlet distribution which is a conjugate prior distribution of the multinomial distribution.

$$\mathbf{z} \sim p(\mathbf{z}) = \text{Dirichlet}(\boldsymbol{\alpha}), \ \mathbf{x} \sim p_\theta(\mathbf{x}|\mathbf{z}) \tag{6}$$

**Inference sub-model.** The probabilistic encoder with an approximating posterior distribution $q_\phi(\mathbf{z}|\mathbf{x})$ is designed to be Dirichlet($\hat{\boldsymbol{\alpha}}$). The approximated posterior parameter $\hat{\boldsymbol{\alpha}}$ is derived by the MLP from the observation **x** with the softplus output function, so the outputs can be positive values constrained by the Dirichlet distribution. Here, we do not directly sample **z** from the Dirichlet distribution. Instead, we use the Gamma composition method described in Section 2.2. Firstly, we draw $\mathbf{v} \sim \text{MultiGamma}(\boldsymbol{\alpha}, \beta \cdot \mathbf{1}_K)$. Afterwards, we normalize **v** with its summation $\sum v_i$.

The objective function to optimize the model parameters, $\theta$ and $\phi$, is composed of Equation (1) and (3). Equation (7) is the loss function to optimize after the composition. The inverse Gamma CDF method explained in the next paragraph enables the backpropagation flows to the input with the stochastic gradient method. Here, for the fair comparison of expressing the Dirichlet distribution between the inverse Gamma CDF approximation method and the softmax Gaussian method, we set $\alpha_k = 1 - 1/K$ when $\mu_k = 0$ and $\Sigma_k = 1$ by using Equation (5); and $\beta = 1$.

$$\mathcal{L}(\mathbf{x}) = \mathbb{E}_{q_{\phi(\mathbf{z}|\mathbf{x})}}[\log p_\theta(\mathbf{x}|\mathbf{z})] - \left(\sum \log \Gamma(\alpha_k) - \sum \log \Gamma(\hat{\alpha}_k) + \sum (\hat{\alpha}_k - \alpha_k)\psi(\hat{\alpha}_k)\right) \quad (7)$$

**Approximation with inverse Gamma CDF.** A previous work Knowles (2015) suggested that, if $X \sim \text{Gamma}(\alpha, \beta)$, and if $F(x; \alpha, \beta)$ is a CDF of the random variable $X$, the inverse CDF can be approximated as $F^{-1}(u; \alpha, \beta) \approx \beta^{-1}(u\alpha\Gamma(\alpha))^{1/\alpha}$. Hence, we can introduce an auxiliary variable $u \sim \text{Uniform}(0, 1)$ to take over all the randomness of $X$, and we treat the Gamma sampled $X$ as a deterministic value in terms of $\alpha$ and $\beta$.

It should be noted that there has been a practice of utilizing the combination of decomposing a Dirichlet distribution and approximating each Gamma component with inverse Gamma CDF. However, such practices have not been examined with its learning properties and applicabilities. The following section shows a new aspect of *component collapsing* that can be remedied by this combination on Dirichlet prior in VAE, and the section illustrates the performance gains in a certain set of applications, i.e. topic modeling.

## 4 EXPERIMENTAL RESULTS

This section reports the experimental results with the following experiment settings: 1) a pure VAE model; 2) a semi-supervised classification task with VAEs; 3) a supervised classification task with VAEs; and 4) topic models with DirVAE augmentations.

### 4.1 EXPERIMENTS FOR REPRESENTATION LEARNING OF VAEs

**Baseline models.** We select the following models as baseline alternatives of the DirVAE: 1) the standard GVAE; 2) the GVAE with softmax (GVAE-Softmax) approximating the Dirichlet distribution with the softmax Gaussian distribution; 3) the SBVAE with the Kumaraswamy distribution (SBVAE-Kuma) & the Gamma composition (SBVAE-Gamma) described in Nalisnick & Smyth (2017); and 4) the DirVAE with the Weibull distribution (DirVAE-Weibull) approximating the Gamma distribution with the Weibull distribution described in Zhang et al. (2018). We use the following benchmark datasets for the experiments: 1) MNIST; 2) MNIST with rotations (MNIST+rot); 3) OMNIGLOT; and 4) SVHN with PCA transformation. We provide the details on the datasets in Appendix D.1.

**Experimental setting.** As a pure VAE model, we compare the DirVAE with the following models: GVAE, GVAE-Softmax, SBVAE-Kuma, SBVAE-Gamma, and DirVAE-Weibull. We use 50-dimension and 100-dimension latent variables for MNIST and OMNIGLOT, respectively. We provide the details of the network structure and optimization in Appendix D.2. We set $\boldsymbol{\alpha} = 0.98 \cdot \mathbf{1}_{50}$ for MNIST and $\boldsymbol{\alpha} = 0.99 \cdot \mathbf{1}_{100}$ for OMNIGLOT for the fair comparison to GVAEs by using Equation (5). All experiments use the Adam optimizer (Kingma & Ba, 2014a) for the parameter learning. Finally, we acknowledge that the hyper-parameter could be updated as Appendix C, and the experiment result with the update is separately reported in Appendix D.2.

**Quantitative result.** For the quantitative comparison among the VAEs, we calculated the Monte-Carlo estimation on the marginal negative log-likelihood, the negative ELBO, and the reconstruction loss. The marginal log-likelihood is approximated as $p(\mathbf{x}) \approx \sum_i \frac{p(\mathbf{x}|\mathbf{z}_i)p(\mathbf{z}_i)}{q(\mathbf{z}_i)}$ for single instance $\mathbf{x}$ where $q(\mathbf{z})$ is a posterior distribution of a prior distribution $p(\mathbf{z})$, which is further derived in Appendix A. Table 1 shows the overall performance of the alternative VAEs. The DirVAE outperforms all baselines in both datasets from the log-likelihood perspective. The value of DirVAE comes from the better encoding of the latent variables that can be used for classification tasks which we examine in the next experiments. While the DirVAE-Weibull follows the prior modeling with the Dirichlet distribution, the Weibull based approximation can be improved by adopting the proposed approach with the inverse Gamma CDF.

Table 1: Negative log-likelihood, negative ELBO, and reconstruction loss of the VAEs for MNIST and OMNIGLOT dataset. The lower values are the better for all measures.

| | MNIST ($K = 50$) | | | OMNIGLOT ($K = 100$) | | |
|---|---|---|---|---|---|---|
| | Neg. LL | Neg. ELBO | Reconst. Loss | Neg. LL | Neg. ELBO | Reconst. Loss |
| GVAE (Nalisnick & Smyth, 2017) | 96.80 | – | – | – | – | – |
| SBVAE-Kuma (Nalisnick & Smyth, 2017) | 98.01 | – | – | – | – | – |
| SBVAE-Gamma (Nalisnick & Smyth, 2017) | 100.74 | – | – | – | – | – |
| GVAE | $94.54_{\pm0.79}$ | $\mathbf{98.58_{\pm0.04}}$ | $\mathbf{74.31_{\pm0.13}}$ | $119.29_{\pm0.44}$ | $126.42_{\pm0.24}$ | $\mathbf{98.90_{\pm0.36}}$ |
| GVAE-Softmax | $98.18_{\pm0.61}$ | $103.49_{\pm0.16}$ | $79.36_{\pm0.82}$ | $130.01_{\pm1.16}$ | $139.73_{\pm0.81}$ | $123.34_{\pm1.43}$ |
| SBVAE-Kuma | $99.27_{\pm0.48}$ | $102.60_{\pm1.81}$ | $83.90_{\pm0.82}$ | $130.73_{\pm2.17}$ | $132.86_{\pm3.03}$ | $119.25_{\pm1.00}$ |
| SBVAE-Gamma | $102.14_{\pm0.69}$ | $135.30_{\pm0.24}$ | $113.89_{\pm0.25}$ | $128.82_{\pm1.82}$ | $149.30_{\pm0.82}$ | $136.36_{\pm1.53}$ |
| DirVAE-Weibull | $114.59_{\pm11.15}$ | $183.33_{\pm2.96}$ | $150.92_{\pm3.70}$ | $140.89_{\pm3.21}$ | $198.01_{\pm2.46}$ | $145.52_{\pm3.13}$ |
| DirVAE | $\mathbf{87.64_{\pm0.64}}$ | $100.47_{\pm0.35}$ | $81.50_{\pm0.27}$ | $\mathbf{108.24_{\pm0.42}}$ | $\mathbf{120.06_{\pm0.35}}$ | $99.78_{\pm0.36}$ |

**Qualitative result.** As a qualitative result, we report the latent dimension-wise reconstructions which are decoder outputs with each one-hot vector in the latent dimension. Figure 3a shows 50 reconstructed images corresponding to each latent dimension from GVAE-Softmax, SBVAE, and DirVAE. We manually ordered the digit-like figures in the ascending order for GVAE-Softmax and DirVAE. We can see that the GVAE-Softmax and the SBVAE have components without significant semantic information, which we will discuss further in Section 4.2, and the DirVAE has interpretable latent dimensions in most of the latent dimensions. Figure 3b also supports the quality of the latent values from DirVAE by visualizing learned latent values through t-SNE (Maaten & Hinton, 2008).

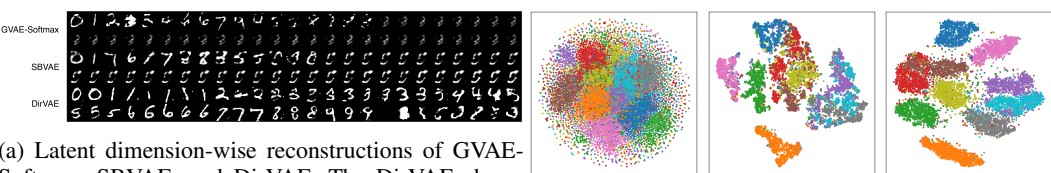

(a) Latent dimension-wise reconstructions of GVAE-Softmax, SBVAE, and DirVAE. The DirVAE shows more meaningful latent dimensions than other VAEs.

(b) (Left) GVAE, (Middle) SBVAE, (Right) DirVAE.

Figure 3: Latent dimension visualization with reconstruction images and t-SNE latent embeddings.

## 4.2 DISCUSSION ON COMPONENT COLLAPSING

**Decoder weight collapsing, a.k.a. component collapsing.** One main issue of GVAE is *component collapsing* that there are a significant number of near-zero decoder weights from the latent neurons to the next decoder neurons. If these weights become near-zero, the values of the latent dimensions loose influence to the next decoder, and this means an inefficient learning given a neural network structure. The same issue occurs when we use the GVAE-Softmax. We rename this component collapsing phenomenon as *decoder weight collapsing* to specifically address the collapsing source.

**Latent value collapsing.** SBVAE claims that SBVAE solved the *decoder weight collapsing* by learning the meaningful weights as shown in Figure 4a. However, we notice that SBVAE produces the output values, not the weight parameters, from the latent dimension to be near-zero in many latent dimensions after averaging many samples obtained from the test dataset. Figure 4b shows the properties of DirVAE and SBVAE from the perspective of the latent value collapsing, which SBVAE shows many near-zero average means and near-zero average variances, while DirVAE does not. The average Fisher kurtosis and average skewness of DirVAE are 5.76 and 2.03, respectively over the dataset, while SBVAE has 20.85 and 4.35, which states that the latent output distribution from SBVAE is more skewed than that of DirVAE. We found out that these near-zero latent values prevent learning on decoder weights, which we introduce as another type of collapsing problem, as *latent value collapsing* that is different from the *decoder weight collapsing*. These results mean that SBVAE distributes the non-near-zero latent values sparsely over a few dimensions while DirVAE samples relatively dense latent values. In other words, DirVAE utilizes the full spectrum of latent dimensions compared to SBVAE, and DirVAE has a better learning capability in the decoder network. Figure 3a supports the argument on the latent value collapsing by activating each and single latent dimension with a one-hot vector through the decoder. The non-changing latent dimension-

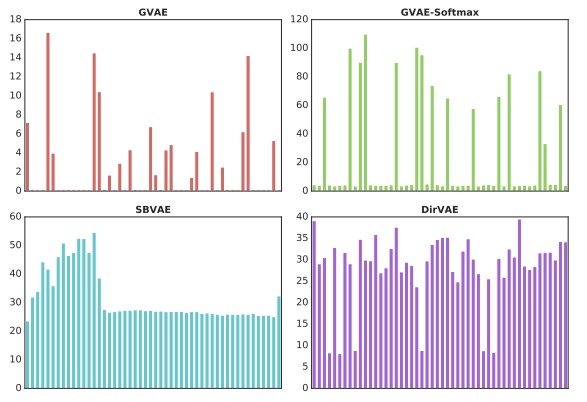
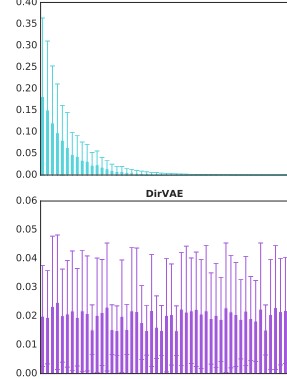

(a) Latent dimension-wise L2-norm of decoder weights of VAEs.  (b) Latent values of VAEs.

Figure 4: Sub-figure 4a shows GVAE and GVAE-Softmax have component collapsing issue, while SBVAE and DirVAE do not. Sub-figure 4b shows that SBVAE has many near-zero output values in the latent dimensions.

wise images of SBVAE proves that there were no generation differences between the two differently activated one-hot latent values.

## 4.3 Application 1. experiments of (semi-)supervised classification with VAEs

**Semi-supervised classification task with VAEs.** There is a previous work demonstrating that the SBVAE outperforms the GVAE in semi-supervised classification task (Nalisnick & Smyth, 2017). The overall model structure for this semi-supervised classification task uses a VAE with separate random variables of $\mathbf{z}$ and $\mathbf{y}$, which is introduced as the *M2* model in the original VAE work (Kingma et al., 2014b). The detailed settings of the semi-supervised classification tasks are enumerated in Appendix D.3. Fundamentally, we applied the same experimental settings to GVAE, SBVAE, and DirVAE in this experiment, as specified by the authors in Nalisnick & Smyth (2017).

Table 2 enumerates the performances of the GVAE, the SBVAE, and the DirVAE, and the result shows that the error rate of classification result using $10\%, 5\%$ and $1\%$ of labeled data for each dataset. In general, the experiment shows that the DirVAE has the best performance out of three alternative VAEs. Also, it should be noted that the performance of the DirVAE is more improved in the most complex task with the SVHN dataset.

Table 2: The error rate of semi-supervised classification task using VAEs.

| | MNIST ($K = 50$) | | | MNIST+rot ($K = 50$) | | | SVHN ($K = 50$) | | |
|---|---|---|---|---|---|---|---|---|---|
| | 10% | 5% | 1% | 10% | 5% | 1% | 10% | 5% | 1% |
| GVAE (Nalisnick & Smyth, 2017) | $\mathbf{3.95}_{\pm 0.15}$ | $\mathbf{4.74}_{\pm 0.43}$ | $11.55_{\pm 2.28}$ | $21.78_{\pm 0.73}$ | $27.72_{\pm 0.69}$ | $38.13_{\pm 0.95}$ | $36.08_{\pm 1.49}$ | $48.75_{\pm 1.47}$ | $69.58_{\pm 1.64}$ |
| SBVAE (Nalisnick & Smyth, 2017) | $4.86_{\pm 0.14}$ | $5.29_{\pm 0.39}$ | $7.34_{\pm 0.47}$ | $11.78_{\pm 0.39}$ | $14.27_{\pm 0.58}$ | $27.67_{\pm 1.39}$ | $32.08_{\pm 4.00}$ | $37.07_{\pm 5.22}$ | $61.37_{\pm 3.60}$ |
| DirVAE | $4.60_{\pm 0.07}$ | $5.05_{\pm 0.18}$ | $\mathbf{7.00}_{\pm 0.17}$ | $\mathbf{11.18}_{\pm 0.32}$ | $\mathbf{13.53}_{\pm 0.46}$ | $\mathbf{26.20}_{\pm 0.66}$ | $\mathbf{24.81}_{\pm 1.13}$ | $\mathbf{28.45}_{\pm 1.14}$ | $\mathbf{55.99}_{\pm 3.30}$ |

**Supervised classification task with latent values of VAEs.** Also, we tested the performance of the supervised classification task with the learned latent representation from the VAEs. We applied the vanilla version of VAEs to the datasets, and we classified the latent representation of instances with $k$-Nearest Neighbor ($k$NN) which is one of the simplest classification algorithms. Hence, this experiment can better distinguish the performance of the representation learning in the classification task. Further experimental details can be found in Appendix D.4.

Table 3 enumerates the performances from the experimented VAEs in the datasets of MNIST and OMNIGLOT. Both datasets indicated that the DirVAE shows the best performance in reducing the classification error, which we conjecture that the performance is gathered from the better representation learning. It should be noted that, to our knowledge, this is the first reported comparison of latent representation learning on VAEs with $k$NN in the supervised classification using OMNIGLOT dataset. We identified that the classification with OMNIGLOT is difficult given that the $k$NN error

rates with the raw original data are as high as 69.94%, 69.41%, and 70.10%. This high error rate mainly originates from the number of classification categories which is 50 categories in our test setting of OMNIGLOT, compared to 10 categories in MNIST.

Table 3: The error rate of $k$NN with the latent representations of VAEs.

| | MNIST ($K = 50$) | | | OMNIGLOT ($K = 100$) | | |
|---|---|---|---|---|---|---|
| | $k = 3$ | $k = 5$ | $k = 10$ | $k = 3$ | $k = 5$ | $k = 10$ |
| GVAE (Nalisnick et al., 2016) | 28.40 | 20.96 | 15.33 | – | – | – |
| SBVAE (Nalisnick et al., 2016) | 9.34 | 8.65 | 8.90 | – | – | – |
| DLGMM (Nalisnick et al., 2016) | 9.14 | 8.38 | 8.42 | – | – | – |
| GVAE | $27.16_{\pm0.48}$ | $20.20_{\pm0.93}$ | $14.89_{\pm0.40}$ | $92.34_{\pm0.25}$ | $91.21_{\pm0.18}$ | $88.79_{\pm0.35}$ |
| GVAE-Softmax | $25.68_{\pm2.64}$ | $21.79_{\pm2.17}$ | $18.75_{\pm2.06}$ | $94.76_{\pm0.20}$ | $94.22_{\pm0.37}$ | $92.98_{\pm0.42}$ |
| SBVAE | $10.01_{\pm0.52}$ | $9.58_{\pm0.47}$ | $9.39_{\pm0.54}$ | $86.90_{\pm0.82}$ | $85.10_{\pm0.89}$ | $82.96_{\pm0.64}$ |
| DirVAE | $\mathbf{5.98}_{\pm\mathbf{0.06}}$ | $\mathbf{5.29}_{\pm\mathbf{0.06}}$ | $\mathbf{5.06}_{\pm\mathbf{0.06}}$ | $\mathbf{76.55}_{\pm\mathbf{0.23}}$ | $\mathbf{73.81}_{\pm\mathbf{0.29}}$ | $\mathbf{70.95}_{\pm\mathbf{0.29}}$ |
| Raw Data | 3.00 | 3.21 | 3.44 | 69.94 | 69.41 | 70.10 |

## 4.4 APPLICATION 2. EXPERIMENTS OF TOPIC MODEL AUGMENTATION WITH DIRVAE

One usefulness of the Dirichlet distribution is being a conjugate prior to the multinomial distribution, so it has been widely used in the field of topic modeling, such as *Latent Dirichlet Allocation* (LDA) (Blei et al., 2003). Recently, some neural variational topic (or document) models have been suggested, for example, ProdLDA (Srivastava & Sutton, 2017), NVDM (Miao et al., 2016), and GSM (Miao et al., 2017). NVDM used the GVAE, and the GSM used the GVAE-Softmax to make the sum-to-one positive topic vectors. Meanwhile, ProdLDA assume the prior distribution to be the Dirichlet distribution with the softmax Laplace approximation. To verify the usefulness of the DirVAE, we replace the probabilistic encoder part of the DirVAE to each model. Two popular performance measures in the topic model fields, which are perplexity and topic coherence via normalized pointwise mutual information (NPMI) (Lau et al., 2014), have been used with *20Newsgroups* and *RCV1-v2* datasets. Further details of the experiments can be found in Appendix D.5. Table 4 indicates that the augmentation of DirVAE improves the performance in general. Additionally, the best performers from the two measurements are always the experiment cell with DirVAE augmentation except for the perplexity of RCV1-v2, which still remains competent.

Table 4: Topic modeling performances of perpexity and NPMI with DirVAE augmentations.

| | | 20Newsgroups ($K = 50$) | | | | RCV1-v2 ($K = 100$) | | | |
|---|---|---|---|---|---|---|---|---|---|
| | | ProdLDA | NVDM | GSM | LDA (Gibbs) | ProdLDA | NVDM | GSM | LDA (Gibbs) |
| Perplexity | Reported | 1172 | 837 | 822 | - | - | - | - | - |
| | Reproduced | $1219_{\pm8.87}$ | $810_{\pm2.60}$ | $954_{\pm1.22}$ | $1314_{\pm18.50}$ | $1190_{\pm45.24}$ | $\mathbf{796}_{\pm\mathbf{6.24}}$ | $1386_{\pm21.06}$ | $1126_{\pm12.66}$ |
| | Add SBVAE | $1164_{\pm2.55}$ | $878_{\pm14.21}$ | $980_{\pm13.50}$ | - | $1077_{\pm22.57}$ | $1050_{\pm12.19}$ | $1670_{\pm4.78}$ | - |
| | Add DirVAE | $1114_{\pm2.30}$ | $\mathbf{752}_{\pm\mathbf{12.17}}$ | $916_{\pm1.64}$ | - | $992_{\pm2.19}$ | $809_{\pm12.60}$ | $1526_{\pm6.11}$ | - |
| NPMI | Reported | 0.240 | 0.186 | 0.121 | - | - | - | - | - |
| | Reproduced | $0.273_{\pm0.019}$ | $0.119_{\pm0.003}$ | $0.199_{\pm0.006}$ | $0.225_{\pm0.002}$ | $0.194_{\pm0.005}$ | $0.023_{\pm0.002}$ | $0.267_{\pm0.019}$ | $0.266_{\pm0.006}$ |
| | Add SBVAE | $0.247_{\pm0.015}$ | $0.162_{\pm0.007}$ | $0.162_{\pm0.006}$ | - | $0.190_{\pm0.006}$ | $0.116_{\pm0.016}$ | $0.207_{\pm0.004}$ | - |
| | Add DirVAE | $\mathbf{0.359}_{\pm\mathbf{0.026}}$ | $0.247_{\pm0.010}$ | $0.201_{\pm0.003}$ | - | $0.193_{\pm0.004}$ | $0.131_{\pm0.015}$ | $\mathbf{0.308}_{\pm\mathbf{0.005}}$ | - |

## 5 CONCLUSION

Recent advances in VAEs have become one of the cornerstones in the field of DGMs. The VAEs infer the parameters of explicitly described latent variables, so the VAEs are easily included in the conventional PGMs. While this merit has motivated the diverse cases of merging the VAEs to the graphical models, we ask the fundamental quality of utilizing the GVAE where many models have latent values to be categorical probabilities. The softmax function cannot reproduce the multi-modal distribution that the Dirichlet distribution can. Recognizing this problem, there have been some previous works that approximated the Dirichlet distribution in the VAE settings by utilizing the Weibull distribution or the softmax Gaussian distribution, but the DirVAE with the inverse Gamma CDF shows the better learning performance in our experiments of the representation: the semi-supervised, the supervised classifications, and the topic models. Moreover, DirVAE shows no component collapsing and it leads to better latent representation and performance gain. The proposed DirVAE can be widely used if we recall the popularity of the conjugate relation between the multinomial and the Dirichlet distributions because the proposed DirVAE can be a brick to the construction of complex probabilistic models with neural networks.

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

APPENDIX

This is an appendix for *Dirichlet Variational Autoencoder*. Here, we describe the derivations of key equations and experimental setting details which were used in the body of the paper. The detailed information such as model names, parameter names, or experiment assumptions is based on the main paper.

## A  MONTE-CARLO ESTIMATION ON THE MARGINAL LIKELIHOOD

**Proposition A.1.** *The marginal log-likelihood is approximated as $p(\mathbf{x}) \approx \sum_i \frac{p(\mathbf{x}|\mathbf{z}_i)p(\mathbf{z}_i)}{q(\mathbf{z}_i)}$, where $q(\mathbf{z})$ is a posterior distribution of a prior distribution $p(\mathbf{z})$.*

*Proof.*

$$p(\mathbf{x}) = \int_{\mathbf{z}} p(\mathbf{x}, \mathbf{z}) d\mathbf{z} = \int_{\mathbf{z}} p(\mathbf{x}, \mathbf{z}) \frac{q(\mathbf{z})}{q(\mathbf{z})} d\mathbf{z}$$

$$= \int_{\mathbf{z}} p(\mathbf{x}|\mathbf{z})p(\mathbf{z}) \frac{q(\mathbf{z})}{q(\mathbf{z})} d\mathbf{z} = \int_{\mathbf{z}} \frac{p(\mathbf{x}|\mathbf{z})p(\mathbf{z})}{q(\mathbf{z})} q(\mathbf{z}) d\mathbf{z}$$

$$\approx \sum_i \frac{p(\mathbf{x}|\mathbf{z}_i)p(\mathbf{z}_i)}{q(\mathbf{z}_i)} \text{ where } \mathbf{z}_i \sim q(\mathbf{z})$$

$\square$

## B  KL DIVERGENCE OF TWO MULTI-GAMMA DISTRIBUTIONS

**Proposition B.1.** *Define $\mathbf{X} = (X_1, \cdots, X_K) \sim MultiGamma(\boldsymbol{\alpha}, \beta \cdot \mathbf{1}_K)$ as a vector of $K$ independent Gamma random variables $X_k \sim Gamma(\alpha_k, \beta)$ where $\alpha_k, \beta > 0$ for $k = 1, \cdots, K$. The KL divergence between two MultiGamma distributions $P = MultiGamma(\boldsymbol{\alpha}, \beta \cdot \mathbf{1}_K)$ and $Q = MultiGamma(\hat{\boldsymbol{\alpha}}, \beta \cdot \mathbf{1}_K)$ can be derived as the following:*

$$KL(Q||P) = \sum \log \Gamma(\alpha_k) - \sum \log \Gamma(\hat{\alpha}_k) + \sum (\hat{\alpha}_k - \alpha_k)\psi(\hat{\alpha}_k) , \tag{8}$$

*where $\psi$ is a digamma function.*

*Proof.* Note that the derivative of a Gamma-like function $\frac{\Gamma(\alpha)}{\beta^\alpha}$ can be derived as follows:

$$\frac{d}{d\alpha} \frac{\Gamma(\alpha)}{\beta^\alpha} = \beta^{-\alpha}(\Gamma'(\alpha) - \Gamma(\alpha) \log \beta) = \int_0^\infty x^{\alpha-1} e^{-\beta x} \log x \ dx .$$

Then, we have the following.

$$\mathrm{KL}(Q||P) = \int_{\mathcal{D}} q(\mathbf{x}) \log \frac{q(\mathbf{x})}{p(\mathbf{x})} \ d\mathbf{x}$$

$$= \int_0^\infty \cdots \int_0^\infty \prod \mathrm{Gamma}(\hat{\alpha}_k, \beta) \log \frac{\beta^{\sum \hat{\alpha}_k} \prod \Gamma^{-1}(\hat{\alpha}_k) e^{-\beta \sum x_k} \prod x_k^{\hat{\alpha}_k-1}}{\beta^{\sum \alpha_k} \prod \Gamma^{-1}(\alpha_k) e^{-\beta \sum x_k} \prod x_k^{\alpha_k-1}} \ d\mathbf{x}$$

$$= \int_0^\infty \cdots \int_0^\infty \prod \mathrm{Gamma}(\hat{\alpha}_k, \beta)$$

$$\times \left[ \sum (\hat{\alpha}_k - \alpha_k) \log \beta + \sum \log \Gamma(\alpha_k) - \sum \log \Gamma(\hat{\alpha}_k) + \sum (\hat{\alpha}_k - \alpha_k) \log x_k \right] d\mathbf{x}$$

$$= \left[ \sum (\hat{\alpha}_k - \alpha_k) \log \beta + \sum \log \Gamma(\alpha_k) - \sum \log \Gamma(\hat{\alpha}_k) \right]$$

$$+ \int_0^\infty \cdots \int_0^\infty \frac{\beta^{\hat{\alpha}_k}}{\prod \Gamma(\hat{\alpha}_k)} e^{-\beta \sum x_k} \prod x_k^{\hat{\alpha}_k-1} \left( \sum (\hat{\alpha}_k - \alpha_k) \log x_k \right) d\mathbf{x}$$

$$= \Big[ \sum (\hat{\alpha}_k - \alpha_k) \log \beta + \sum \log \Gamma(\alpha_k) - \sum \log \Gamma(\hat{\alpha}_k) \Big]$$

$$+ \sum (\hat{\alpha}_k - \alpha_k) \beta^{\hat{\alpha}_k} \Gamma^{-1}(\hat{\alpha}_k) \beta^{-\hat{\alpha}_k} \big( \Gamma'(\hat{\alpha}_k) - \Gamma(\hat{\alpha}_k) \log \beta \big)$$

$$= \sum (\hat{\alpha}_k - \alpha_k) \log \beta + \sum \log \Gamma(\alpha_k) - \sum \log \Gamma(\hat{\alpha}_k) + \sum (\hat{\alpha}_k - \alpha_k)(\psi(\hat{\alpha}_k) - \log \beta)$$

$$= \sum \log \Gamma(\alpha_k) - \sum \log \Gamma(\hat{\alpha}_k) + \sum (\hat{\alpha}_k - \alpha_k)\psi(\hat{\alpha}_k)$$

$\square$

## C  HYPER-PARAMETER $\alpha$ LEARNING STRATEGY

In this section, we introduce the method of moment estimator (MME) to update the Dirichlet prior parameter $\alpha$. Suppose we have a set of sum-to-one proportions $\mathcal{D} = \{\mathbf{p}_1, \cdots, \mathbf{p}_N\}$ sampled from Dirichlet($\alpha$), then the MME update rule is as the following:

$$\alpha_k \leftarrow \frac{S}{N} \sum_n p_{n,k} \text{ where } S = \frac{1}{K} \sum_k \frac{\tilde{\mu}_{1,k} - \tilde{\mu}_{2,k}}{\tilde{\mu}_{2,k} - \tilde{\mu}_{1,k}^2} \text{ for } \tilde{\mu}_{j,k} = \frac{1}{N} \sum_n p_{n,k}^j . \tag{9}$$

After the burn-in period for stabilizing the neural network parameters, we use the MME for the hyper-parameter learning using the sampled latent values during training. We alternatively update the neural network parameters and hyper-parameter $\alpha$. We choose this estimator because of its closed form nature and consistency (Minka, 2000). The usefulness of the hyper-parameter update can be found in Appendix D.2.

**Proposition C.1.** *Given a proportion set $\mathcal{D} = \{\mathbf{p}_1, \cdots, \mathbf{p}_N\}$ sampled from Dirichlet($\alpha$), MME of the hyper-parameter $\alpha$ is as the following:*

$$\alpha_k \leftarrow \frac{S}{N} \sum_n p_{n,k} \text{ where } S = \frac{1}{K} \sum_k \frac{\tilde{\mu}_{1,k} - \tilde{\mu}_{2,k}}{\tilde{\mu}_{2,k} - \tilde{\mu}_{1,k}^2} \text{ for } \tilde{\mu}_{j,k} = \frac{1}{N} \sum_n p_{n,k}^j .$$

*Proof.* Define $\mu_{j,k} = \mathbb{E}[p_k^j]$ as the $j^{\text{th}}$ moment of the $k^{\text{th}}$ dimension of Dirichlet distribution with prior $\alpha$. Then, by the law of large number, $\mu_{j,k} \approx \tilde{\mu}_{j,k}$. It can be easily shown that $\mu_{1,k} = \frac{\alpha_k}{\sum_i \alpha_i}$ and $\mu_{2,k} = \frac{\alpha_k}{\sum_i \alpha_i} \frac{1+\alpha_k}{1+\sum_i \alpha_i} = \mu_{1,k} \frac{1+\alpha_k}{1+\sum_i \alpha_i}$ so that

$$\text{numerator}\Big( \frac{\mu_{1,k} - \mu_{2,k}}{\mu_{2,k} - \mu_{1,k}^2} \Big) = \frac{\alpha_k}{\sum_i \alpha_i} - \frac{\alpha_k}{\sum_i \alpha_i} \frac{1+\alpha_k}{1+\sum_i \alpha_i}$$

$$= \frac{\alpha_k(\sum_{i \neq k} \alpha_i)}{(\sum_i \alpha_i)(1+\sum_i \alpha_i)}$$

$$\text{denominator}\Big( \frac{\mu_{1,k} - \mu_{2,k}}{\mu_{2,k} - \mu_{1,k}^2} \Big) = \frac{\alpha_k}{\sum_i \alpha_i} \frac{1+\alpha_k}{1+\sum_i \alpha_i} - \Big( \frac{\alpha_k}{\sum_i \alpha_i} \Big)^2$$

$$= \frac{\alpha_k(\sum_{i \neq k} \alpha_i)}{(\sum_i \alpha_i)^2(1+\sum_i \alpha_i)}$$

holds for each $k = 1, \cdots, K$. Therefore,

$$\sum_i \alpha_i = \frac{\mu_{1,k} - \mu_{2,k}}{\mu_{2,k} - \mu_{1,k}^2} \approx \frac{1}{K} \sum_k \frac{\mu_{1,k} - \mu_{2,k}}{\mu_{2,k} - \mu_{1,k}^2} \approx \frac{1}{K} \sum_k \frac{\tilde{\mu}_{1,k} - \tilde{\mu}_{2,k}}{\tilde{\mu}_{2,k} - \tilde{\mu}_{1,k}^2}$$

and hence,

$$\hat{\alpha}_k = (\sum_i \alpha_i)\tilde{\mu}_{1,k} = \frac{S}{N} \sum_n p_{n,k}.$$

$\square$

## D EXPERIMENTAL SETTINGS

In this section, we support Section 4 in the original paper with more detailed experimental settings. Our Tensorflow implementation is available at `https://TO_BE_RELEASED`.

### D.1 DATASET DESCRIPTION

We use the following benchmark datasets for the experiments in the original paper: 1) MNIST; 2) MNIST with rotations (MNIST+rot); 3) OMNIGLOT; and 4) SVHN with PCA transformation. MNIST (LeCun et al., 1998) is a hand-written digit image dataset of size $28 \times 28$ with 10 labels, consists of $60,000$ training data and $10,000$ testing data. MNIST+rot data is reproduced by the authors of Nalisnick & Smyth (2017) consists of MNIST and rotated MNIST[1]. OMNIGLOT[2] (Lake et al., 2013; Snderby et al., 2016) is another hand-written image dataset of characters with $28 \times 28$ size and 50 labels, consists of $24,345$ training data and $8,070$ testing data. SVHN[3] is a Street View House Numbers image dataset with the dimension-reduction by PCA into $500$ dimensions (Nalisnick & Smyth, 2017).

### D.2 REPRESENTATION LEARNING OF VAEs

We divided the datasets into {train,valid,test} as the following: MNIST $= \{45,000 : 5,000 : 10,000\}$ and OMNIGLOT $= \{22,095 : 2,250 : 8,070\}$.

For MNIST, we use 50-dimension latent variables with two hidden layers in the encoder and one hidden layer in the decoder of $500$ dimensions. We set $\boldsymbol{\alpha} = 0.98 \cdot \mathbf{1}_{50}$ for the fair comparison to GVAEs using the Equation (5). The batch size was set to be 100. For OMNIGLOT, we use 100-dimension latent variables with two hidden layers in the encoder and one hidden layer in the decoder of $500$ dimensions. We assume $\boldsymbol{\alpha} = 0.99 \cdot \mathbf{1}_{100}$ for the fair comparison to the GVAEs using the Equation (5). The batch size was set to be 15.

For both datasets, the gradient clipping is used; ReLU function (Nair & Hinton, 2010) is used as an activation function in hidden layers; Xavier initialization (Glorot & Bengio, 2010) is used for the neural network parameter initialization; and the Adam optimizer (Kingma & Ba, 2014a) is used as an optimizer with learning rate `5e-4` for all VAEs except `3e-4` for the SBVAEs. The prior assumptions for each VAE is the following: 1) $\mathcal{N}(\mathbf{0}, \mathbf{I})$ for the GVAE and the GVAE-Softmax; 2) GEM(5) for the SBVAEs; and 3) Dirichlet($0.98 \cdot \mathbf{1}_{50}$) (MNIST) and Dirichlet($0.99 \cdot \mathbf{1}_{100}$) (OMNIGLOT) for the DirVAE-Weibull. Finally, to compute the marginal log-likelihood, we used 100 samples for each $1,000$ randomly selected from the test data.

We add the result of VAE with 20 normalizing flows (GVAE-NF20) (Rezende & Mohamed, 2015) as a baseline in Table 5. Also, latent dimension-wise decoder weight norm and t-SNE visualization on latent embeddings of MNIST is given in Figure 5a and 5b which correspond to Figure 4a and 3, respectively.

Additionally, DirVAE-Learning use the same $\boldsymbol{\alpha}$ for the initial value, but the DirVAE-Learning optimizes hyper-parameter $\boldsymbol{\alpha}$ by the following stages through the learning iterations using the MME method in Appendix C: 1) the burn-in period for stabilizing the neural network parameters; 2) the alternative update period for the neural network parameters and $\boldsymbol{\alpha}$; and 3) the update period for the neural network parameters with the fixed learned hyper-parameter $\boldsymbol{\alpha}$. Table 5 shows that there are improvements in the marginal log-likelihood, ELBO, and reconstruction loss with DirVAE-Learning in both datasets. We also give the learned hyper-parameter $\boldsymbol{\alpha}$ in Figure 6.

### D.3 SEMI-SUPERVISED CLASSIFICATION TASK WITH VAEs

The overall model structure for this semi-supervised classification task uses a VAE with a separate random variable of $\mathbf{z}$ and $\mathbf{y}$, which is introduced as the *M2* model in the original VAE work (Kingma et al., 2014b). However, the same task with the SBVAE uses a different model modified to ignore

---

[1] `http://www.iro.umontreal.ca/ lisa/twiki/bin/view.cgi/Public/MnistVariations`
[2] `https://github.com/yburda/iwae/tree/master/datasets/OMNIGLOT`
[3] `http://ufldl.stanford.edu/housenumbers/`

Table 5: Negative log-likelihood, negative ELBO, and reconstruction loss of the VAEs for MNIST and OMNIGLOT dataset. The lower values are the better for all measures.

| | MNIST ($K = 50$) | | | OMNIGLOT ($K = 100$) | | |
| --- | --- | --- | --- | --- | --- | --- |
| | Neg. LL | Neg. ELBO | Reconst. Loss | Neg. LL | Neg. ELBO | Reconst. Loss |
| GVAE (Nalisnick & Smyth, 2017) | 96.80 | – | – | – | – | – |
| SBVAE-Kuma (Nalisnick & Smyth, 2017) | 98.01 | – | – | – | – | – |
| SBVAE-Gamma (Nalisnick & Smyth, 2017) | 100.74 | – | – | – | – | – |
| GVAE | $94.54_{\pm 0.79}$ | $\mathbf{98.58}_{\pm \mathbf{0.04}}$ | $\mathbf{74.31}_{\pm \mathbf{0.13}}$ | $119.29_{\pm 0.44}$ | $126.42_{\pm 0.24}$ | $\mathbf{98.90}_{\pm \mathbf{0.36}}$ |
| GVAE-Softmax | $98.18_{\pm 0.61}$ | $103.49_{\pm 0.16}$ | $79.36_{\pm 0.82}$ | $130.01_{\pm 1.16}$ | $139.73_{\pm 0.81}$ | $123.34_{\pm 1.43}$ |
| GVAE-NF20 | $95.87_{\pm 0.64}$ | $113.14_{\pm 0.47}$ | $90.09_{\pm 1.19}$ | $113.51_{\pm 1.29}$ | $129.82_{\pm 0.64}$ | $108.96_{\pm 1.19}$ |
| SBVAE-Kuma | $99.27_{\pm 0.48}$ | $102.60_{\pm 1.81}$ | $83.90_{\pm 0.82}$ | $130.73_{\pm 2.17}$ | $132.86_{\pm 3.03}$ | $119.25_{\pm 1.00}$ |
| SBVAE-Gamma | $102.14_{\pm 0.69}$ | $135.30_{\pm 0.24}$ | $113.89_{\pm 0.25}$ | $128.82_{\pm 1.82}$ | $149.30_{\pm 0.82}$ | $136.36_{\pm 1.53}$ |
| DirVAE-Weibull | $114.59_{\pm 11.15}$ | $183.33_{\pm 2.96}$ | $150.92_{\pm 3.70}$ | $140.89_{\pm 3.21}$ | $198.01_{\pm 2.46}$ | $145.52_{\pm 3.13}$ |
| DirVAE | $\mathbf{87.64}_{\pm \mathbf{0.64}}$ | $100.47_{\pm 0.35}$ | $81.50_{\pm 0.27}$ | $\mathbf{108.24}_{\pm \mathbf{0.42}}$ | $\mathbf{120.06}_{\pm \mathbf{0.35}}$ | $99.78_{\pm 0.36}$ |
| DirVAE-Learning | $\mathbf{84.42}_{\pm \mathbf{0.53}}$ | $99.88_{\pm 0.40}$ | $80.73_{\pm 0.31}$ | $\mathbf{100.01}_{\pm \mathbf{0.52}}$ | $119.73_{\pm 0.31}$ | $99.55_{\pm 0.32}$ |

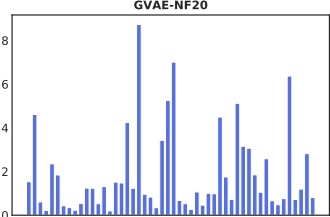

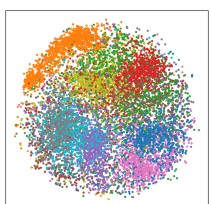

(a) Latent dimension-wise L2-norm of decoder weights of GVAE-NF20.

(b) GVAE-NF20 t-SNE visualization.

Figure 5: Decoder weight collapsing and t-SNE latent embeddings visualization of GVAE-NF20 on MNIST.

the relation between the class label variable $\mathbf{y}$ and the latent variable $\mathbf{z}$, but they still share the same parent nodes: $q_\phi(\mathbf{z}, \mathbf{y}|\mathbf{x}) = q_\phi(\mathbf{z}|\mathbf{x})q_\phi(\mathbf{y}|\mathbf{x})$ where $q_\phi(\mathbf{y}|\mathbf{x})$ is a discrimitive network for the unseen labels. We follow the structure of SBVAE. Finally, the below are the objective functions to optimize for the labeled and the unlabeled instances of the semi-supervised classification task, respectively:

$$\log p(\mathbf{x}, \mathbf{y}) \geq \mathcal{L}_{\text{labeled}}(\mathbf{x}, \mathbf{y}) = \mathbb{E}_{q_{\phi(\mathbf{z}|\mathbf{x})}}[\log p_\theta(\mathbf{x}|\mathbf{z}, \mathbf{y})] - \text{KL}(q_\phi(\mathbf{z}|\mathbf{x})||p_\theta(\mathbf{z})) + \log q_\phi(\mathbf{y}|\mathbf{x}) , \quad (10)$$

$$\log p(\mathbf{x}) \geq \mathcal{L}_{\text{unlabeled}}(\mathbf{x}) = \mathbb{E}_{q_\phi(\mathbf{z}, \mathbf{y}|\mathbf{x})}[\log p_\theta(\mathbf{x}|\mathbf{z}, \mathbf{y}) + \mathbb{H}(q_\phi(\mathbf{y}|\mathbf{x}))] - \text{KL}(q_\phi(\mathbf{z}|\mathbf{x})||p_\theta(\mathbf{z})) . \quad (11)$$

In the above, $\mathbb{H}$ is an entropy function. The actual training on the semi-supervised learning optimizes the weighted sum of Equation (10) and (11) with a ratio hyper-parameter $0 < \lambda < 1$.

The datasets are divided into {train, valid, test} as the following: MNIST $= \{45,000 : 5,000 : 10,000\}$, MNIST+rot $= \{70,000 : 10,000 : 20,000\}$, and SVHN $= \{65,000 : 8,257 : 26,032\}$. For SVHN, dimension reduction into $500$ dimensions by PCA is applied as preprocessing.

Fundamentally, we applied the same experimental settings to GVAE, SBVAE and DirVAE in this experiment, as specified by the authors in Nalisnick & Smyth (2017).[4,5] Specifically, the three VAEs used the same network structures of 1) a hidden layer of $500$ dimension for MNIST; and 2) four hidden layers of $500$ dimensions for MNIST+rot and SVHN with the residual network for the last three hidden layers. The latent variables have $50$ dimensions for all settings. The ratio parameter $\lambda$ is set to be $0.375$ for the MNISTs, and $0.45$ for SVHN. ReLU function is used as an activation function in hidden layers, and the neural network parameters were initialized by sampling from $\mathcal{N}(0, 0.001)$. The Adam optimizer is used with learning rate $\texttt{3e-4}$ and the batch size was set to be $100$. Finally, the DirVAE sets $\boldsymbol{\alpha} = 0.98 \cdot \mathbf{1}_{50}$ by using Equation (5).

---

[4]https://github.com/enalisnick/stick-breaking_dgms
[5]https://www.ics.uci.edu/~enalisni/sb_dgm_supp_mat.pdf

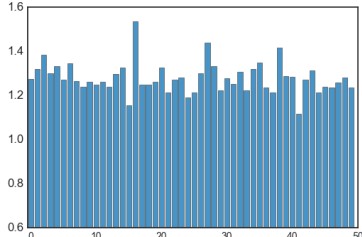

Figure 6: The optimized dimension-wise $\alpha$ values from DirVAE-Learning with MNIST.

## D.4 SUPERVISED CLASSIFICATION TASK WITH LATENT VALUES OF VAEs

For the supervised classification task on the latent representation of the VAEs, we used exactly the same experimental settings as in D.2. Since DLGMM is basically a Gaussian mixture model with the SBVAE, DLGMM is a more complex model than the VAE alternatives. We only report the authors' result from Nalisnick et al. (2016) for the comparison purposes. Additionally, we omit the comparison with the VaDE (Jiang et al., 2017) because the VaDE is more customized to be a clustering model rather than the ordinary VAEs that we choose as baselines.

Table 6: The error rate of $k$NN with the latent representations of VAEs.

|  | MNIST ($K = 50$) | | | OMNIGLOT ($K = 100$) | | |
|---|---|---|---|---|---|---|
|  | $k = 3$ | $k = 5$ | $k = 10$ | $k = 3$ | $k = 5$ | $k = 10$ |
| GVAE (Nalisnick et al., 2016) | 28.40 | 20.96 | 15.33 | – | – | – |
| SBVAE (Nalisnick et al., 2016) | 9.34 | 8.65 | 8.90 | – | – | – |
| DLGMM (Nalisnick et al., 2016) | 9.14 | 8.38 | 8.42 | – | – | – |
| GVAE | $27.16_{\pm 0.48}$ | $20.20_{\pm 0.93}$ | $14.89_{\pm 0.40}$ | $92.34_{\pm 0.25}$ | $91.21_{\pm 0.18}$ | $88.79_{\pm 0.35}$ |
| GVAE-Softmax | $25.68_{\pm 2.64}$ | $21.79_{\pm 2.17}$ | $18.75_{\pm 2.06}$ | $94.76_{\pm 0.20}$ | $94.22_{\pm 0.37}$ | $92.98_{\pm 0.42}$ |
| GVAE-NF20 | $25.72_{\pm 1.58}$ | $20.15_{\pm 1.25}$ | $15.87_{\pm 0.74}$ | $91.25_{\pm 0.12}$ | $90.03_{\pm 0.20}$ | $87.73_{\pm 0.38}$ |
| SBVAE | $10.01_{\pm 0.52}$ | $9.58_{\pm 0.47}$ | $9.39_{\pm 0.54}$ | $86.90_{\pm 0.82}$ | $85.10_{\pm 0.89}$ | $82.96_{\pm 0.64}$ |
| DirVAE | $\mathbf{5.98_{\pm 0.06}}$ | $\mathbf{5.29_{\pm 0.06}}$ | $\mathbf{5.06_{\pm 0.06}}$ | $\mathbf{76.55_{\pm 0.23}}$ | $\mathbf{73.81_{\pm 0.29}}$ | $\mathbf{70.95_{\pm 0.29}}$ |
| Raw Data | 3.00 | 3.21 | 3.44 | 69.94 | 69.41 | 70.10 |

## D.5 TOPIC MODEL AUGMENTATION WITH DIRVAE

For the topic model augmentation experiment, two popular performance measures in the topic model fields, which are perplexity and topic coherence via normalized pointwise mutual information (NPMI) (Lau et al., 2014), have been used with *20Newsgroups*[6] and *RCV1-v2*[7] datasets. 20Newsgroups has $11,258$ train data and $7,487$ test data with vocabulary size $1,995$. For the RCV1-v2 dataset, due to the massive size of the whole data, we randomly sampled $20,000$ train data and $10,000$ test data with vocabulary size $10,000$. The lower is better for the perplexity, and the higher is better for the NPMI.

The specific model structures can be found in the original papers, Srivastava & Sutton (2017); Miao et al. (2016; 2017). We replace the model prior to that of DirVAE to each model and search the hyper-parameter as Table 7 with $1,000$ randomly selected test data. We use 500-dimension hidden layers and 50 topics for 20Newsgroups, and $1,000$-dimension hidden layers and 100 topics for RCV1-v2.

Table 8 shows top-10 high probability words per topic by activating single latent dimensions in the case of 20Newsgroups. Also, we visualized the latent embeddings of documents by t-SNE in Figure 7,8, and 9.

---

[6]`https://github.com/akashgit/autoencoding_vi_for_topic_models`
[7]`http://scikit-learn.org/stable/datasets/rcv1.html`

Table 7: Hyper-parameter selections for DirVAE augmentations.

| | 20Newsgroups ($K = 50$) | | | RCV1-v2 ($K = 100$) | | |
|---|---|---|---|---|---|---|
| | ProdLDA | NVDM | GSM | ProdLDA | NVDM | GSM |
| Add DirVAE | $0.98 \cdot \mathbf{1}_{50}$ | $0.95 \cdot \mathbf{1}_{50}$ | $0.20 \cdot \mathbf{1}_{50}$ | $0.99 \cdot \mathbf{1}_{100}$ | $0.90 \cdot \mathbf{1}_{100}$ | $0.01 \cdot \mathbf{1}_{100}$ |

Table 8: Sample of learned per topic top-10 high probability words from 20Newsgroups with DirVAE augmentation by activating single latent dimensions.

### ProdLDA+DirVAE

| | |
|---|---|
| Topic 1 | turks turkish armenian genocide village armenia armenians muslims turkey greece |
| Topic 2 | doctrine jesus god faith christ scripture belief eternal holy bible |
| Topic 3 | season defensive puck playoff coach score flyers nhl team ice |
| Topic 4 | pitcher braves hitter coach pen defensive injury roger pitch player |
| Topic 5 | ide scsi scsus controller motherboard isa cache mb floppy ram |
| Topic 6 | toolkit widget workstation xlib jpeg xt vendor colormap interface pixel |
| Topic 7 | spacecraft satellite solar shuttle nasa mission professor lunar orbit rocket |
| Topic 8 | knife handgun assault homicide batf criminal gun firearm police apartment |
| Topic 9 | enforcement privacy encrypt encryption ripem wiretap rsa cipher cryptography escrow |
| Topic 10 | min detroit tor det calgary rangers leafs montreal philadelphia cal |

(a) DirVAE augmentation to ProdLDA

### NVDM+DirVAE

| | |
|---|---|
| Topic 1 | armenian azerbaijan armenia genocide armenians turkish militia massacre village turks |
| Topic 2 | arab arabs israeli palestinian jews soldier turks nazi massacre jew |
| Topic 3 | resurrection bible christianity doctrine scripture eternal belief christian faith jesus |
| Topic 4 | hitter season braves pitcher baseball pitch game player defensive team |
| Topic 5 | directory file compile variable update ftp version site copy host |
| Topic 6 | performance speed faster mhz rate clock processor average twice fast |
| Topic 7 | windows microsoft driver dos nt graphic vga card virtual upgrade |
| Topic 8 | seat gear rear tire honda oil front mile wheel engine |
| Topic 9 | patient disease doctor treatment symptom medical health hospital pain medicine |
| Topic 10 | pt la det tor pit pp vs van cal nj |

(b) DirVAE augmentation to NVDM

### GSM+DirVAE

| | |
|---|---|
| Topic 1 | turkish armenian armenians people one turkey armenia turks greek history |
| Topic 2 | israel israeli jews attack world jewish article arab peace land |
| Topic 3 | god jesus christian religion truth believe bible church christ belief |
| Topic 4 | team play game hockey nhl score first division go win |
| Topic 5 | drive video mac card port pc system modem memory speed |
| Topic 6 | image software file version server program system ftp package support |
| Topic 7 | space launch orbit earth nasa moon satellite mission project center |
| Topic 8 | law state gun government right rights case court police crime |
| Topic 9 | price sell new sale offer pay buy good condition money |
| Topic 10 | internet mail computer send list fax phone email address information |

(c) DirVAE augmentation to GSM

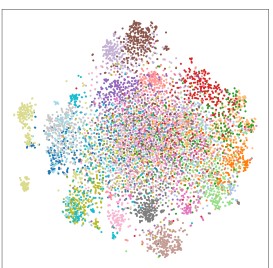 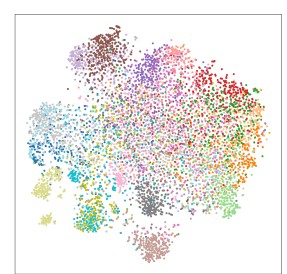 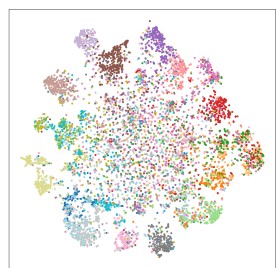

Figure 7: 20Newsgroups latent document embedding visulaization with t-SNE by replacing the model prior to the Dirichlet. (Left) ProdLDA+DirVAE, (Middle) NVDM+DirVAE, (Right) GSM+DirVAE.

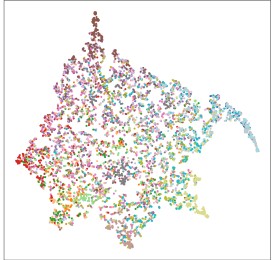 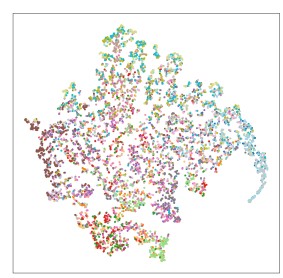 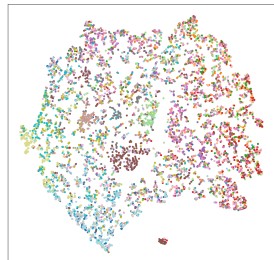

Figure 8: 20Newsgroups latent document embedding visulaization with t-SNE by replacing the model prior to the Stick-Breaking. (Left) ProdLDA+SBVAE, (Middle) NVDM+SBVAE, (Right) GSM+SBVAE.

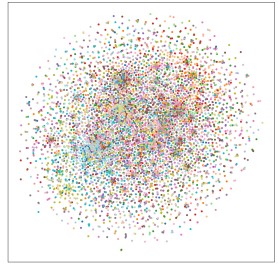 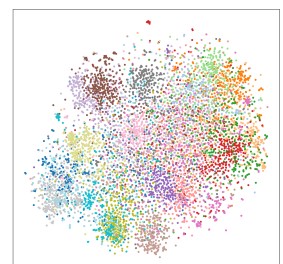 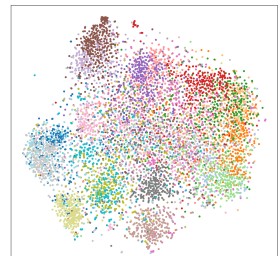

Figure 9: 20Newsgroups latent document embedding visulaization with t-SNE of original models. (Left) ProdLDA, (Middle) NVDM, (Right) GSM.

