# OpenReview forum: "Dirichlet Variational Autoencoder"
_ICLR.cc/2019/Conference_

### Official Review · AnonReviewer1 · 2018-10-18
**A simple method giving improved results**

**Rating:** 7
**Confidence:** 3

**Review:**

Review:

This paper proposes to change the typical Gaussian posterior distribution (and prior) for the latent features z associated to an image x that is used in Variational Autoencoders by a Dirichlet distribution. The work improves over previous attempts based on a soft-max + Gaussian distribution and the soft-max + Weibull distribution. The trick proposed to make feasible training the model includes approximating the inverse CDF of the gamma distribution and using the fact that the Dirichlet distribution can also be obtained as a normalized sum of gamma random variables. The method is compared in several problems. Some analysis of the reasons why it performs better is also carried out.

Quality:

	I think the quality of the paper is high. It is a well written paper in which the choices made are well supported. It also has a strong experimental section.

Clarity:

	The paper is well written and reads very smoothly. I have missed however a more clear statement in the introduction supporting the use of the Dirichlet for the prior and posterior of the latent variables, simply because it seems to give better results and the typical Gaussian choice.

Originality:

	The paper is based on ideas already known. E.g., Dirichlet a normalized sum of gamma random variables and approximation of the inverse CDF of the gamma random variable. The combination of these two techniques is however novel.

Significance:

	The results obtained indicate that the proposed approach improves over previous work on the Dirichlet VAE and on the Gaussian VAE. So I believe the significance of the paper is high.

pros:

	- Good results.

	- Simple method proposed.

	- Extensive experiments.

	- Well written paper.

cons:

	- The idea is a combination of already known techniques put in practice for the VAE.

	- A better motivation that the Dirichlet VAE gives good results should be given at the introduction.

---

> ### Author Response · Authors · 2018-11-11
> **Responses to Reviewer1.**
>
> Thank you for your review.
>
> Firstly, as a motivation for the better result, we can state as the following, and if you are okay with the below sentences, we would like to add it to the introduction part.
> "Due to the component collapsing issues, the existing VAEs have less meaningful latent values or could not effectively use its latent representation. Meanwhile, DirVAE does not have component collapsing due to the multi-modal prior which possibly leads to superior qualitative and quantitative performances. We experimentally showed that the DirVAE has more meaningful or disentangled latent representation by image generation and latent value visualizations."
>
> Secondly, although the techniques are already known, we've rather wanted to focus our paper on the characteristic of Dirichlet prior on VAE such as better latent representation due to no component collapsing, or its applicability like topic modeling.
>
> Thanks again for your valuable comments.
>
> Sincerely.

---

> > ### Comment · AnonReviewer1 · 2018-11-21
> > **Response to author feedback.**
> >
> > Thank you for the response. The sentence looks appropriate.

---

### Official Review · AnonReviewer2 · 2018-10-31
**Limited novelty**

**Rating:** 5
**Confidence:** 5

**Review:**

This paper proposes DirVAE, a variational autoencoder with Dirichlet prior on latent variables. The advantage of using Dirichlet distribution is that due the nature of Dirichlet distribution the model does not suffer from decoder weight collapsing and latent value collapsing. Stochastic gradient variational Bayes with inverse CDF reparametrization of gamma distribution is presented.

The motivation behind using Dirichlet instead of GEM makes sense, but other than that I fail to find any novelty in the paper. The authors should tone down the statement "to our knowledge, combining the two statistical results is the first finding in the machine learning field". Even though left unpublished, I've been using this combination of inverse CDF gamma reparametrization and transformation to Dirichlet all the time for my own problems. It's just trivial once we have both techniques. See also [2], where an improved way of reparametrizing gamma and Dirichlet distribution is presented. The observation that DirVAE does not suffer from latent value collapsing is interesting, but not really surprising.

Minor question
- What is the difference between negative LL's and reconstruction losses in experiments?
- The approximation for inverse CDF of gamma works well only when alpha << 1. How did you treat the regime alpha > 1?


References
[1] Diederik P Kingma, Max Welling, Auto-encoding variational Bayes, ICLR, 2014.
[2] Michael Figurnov, Shakir Mohamed, Andriy Mnih, Implicit reparametrization gradients, arXiv, 2018.

---

> ### Author Response · Authors · 2018-11-12
> **Responses to Reviewer2.**
>
> Thank you for your review.
>
> Firstly, we would like to say thank you for introducing the paper [2]. Even though the paper [2] and our paper lie on the same path in terms of reparametrizing Gamma distribution, paper [2] deals with a general reparametrization trick on various probabilistic distributions while our paper focuses on the advantages and the applicabilities of Dirichlet prior in VAE. We believe that our contribution is not based on reparametrizing Gamma or Dirichlet distiribution, but introducing Dirichlet prior, which is a conjugate multi-modal prior of categorical distribution, on VAE which has better learned latent representation due to no component collapsing. To support this, we did extensive experiments including topic modeling experiments and experimentally showed that DirVAE with the Dirichlet prior does not have component collapsing for the first time in this field. This component collapsing was not experimented and discussed in the prior work of [2]. Moreover, our experiments on the topic modeling shows the consistent performance increases when we apply the DirVAE, which was not discusses in [2].
>
> The below is the response to your questions.
> The log likelihood (LL) is the log-probability that a learner optimizes to train a model given an observed dataset. However, since the likelihood or log-likelihood function is intractable given a latent variable in VAE, so we use the evidence lower bound (ELBO) to optimize the LL. ELBO is a tractable alternative of LL, so the optimization on ELBO is feasible. ELBO term consists of two parts: Reconstruction Error (or Reconstruction Loss, which you asked) and KL divergence terms. Here, Reconstruction Error measures the error between the input and the output, which is an auto-encoder reconstructed input. Additionally, for your information, equation (1) in our paper, can be re-written as follows: Negative Log-likelihood <= Negative ELBO = Reconstruction Loss + KL Divergence.
>
> The author of paper [3] on the inverse Gamma recommends a finite difference approximation method when alpha>1. We only encountered such alpha>1 cases when we updated alphas, and the topic modeling often sets the alpha to be in the range of [0,1]. In the cases of alpha>1, we approached this problem via approximating the inverse function of the Gamma CDF with a Newton method, but the learning performance was not satisfactory. Thus, we left the updated alpha parameters with values greater than one in the appendix.
>
> Sincerely.
>
> References
> [1] Diederik P Kingma, Max Welling, Auto-encoding variational Bayes, ICLR, 2014.
> [2] Michael Figurnov, Shakir Mohamed, Andriy Mnih, Implicit reparametrization gradients, arXiv, 2018.
> [3] David. A. Knowles. Stochastic gradient variational bayes for gamma approximating distributions. arXiv, 2015.

---

> > ### Comment · AnonReviewer2 · 2018-11-21
> > **Thanks for the feedback!**
> >
> > I appreciate your clarifications. My concern about novelty still remains, but I agree that the proper application of well known technique leading to notable performance gains deserve a contribution, so won't argue for the reject if other reviewers vote for the acceptance. As in my initial reveiw, I suggest you to tone down the statement "to our knowledge, combining the two statistical results is the first finding in the machine learning field".

---

> > > ### Author Response · Authors · 2018-11-24
> > > **Response to Reviewer2.**
> > >
> > > We agree that we should tone-down the statement that you've mentioned.
> > >
> > > We re-write the paragraph as following: "It should be noted that there has been a practice of utilizing the combination of decomposing a Dirichlet distribution and approximating each Gamma component with inverse Gamma CDF. However, such practices have not been examined with its learning properties and applicabilities. The following section shows a new aspect of component collapsing that can be remedied by this combination on Dirichlet prior in VAE, and the section illustrates the performance gains in a certain set of applications, i.e. topic modeling."
> > >
> > > Thanks again for your valuable comment.
> > >
> > > Sincerely.

---

### Official Review · AnonReviewer3 · 2018-11-03
**well written paper with novelty concerns**

**Rating:** 6
**Confidence:** 4

**Review:**

In this paper, authors proposes an algorithm to use Dirichlet prior on the variational auto-encoder (VAE). They used this prior as natural conjugate to likelihood distributtion of multinomial (categorical). The paper proposes a way to use scalability power of VAE for data distributed by categorical distribution. In order to apply reparametrization trick, authors have used iid Gamma random variable to construct draw from Dirichlet distribution and have used approximation with inverse gamma CDF,  it is discussed how this method has better performance than other approximations method for gamma distribution such as Weibull and logistic Gaussian.

Authors pointed out, one of the weak points in competing models such as  Guassian softmax prior or Griffith -Engen-McCloskey prior which has been used for Stick breaking VAE is to not encouraging of having multi-modal posteriori, while this prior empower having multi-modal posteriori distribution which give them advantage over previous papers.

 In experimental results, paper has used different datasets of MNIST, MNIST+rotation , OMNIGLOT , 20newsgroup and RCVI and used different measures to compare the existing method with the baselines.

To summarize the contribution of this paper, following three points can be named as main contribution of this paper:
- proposed a Dirichlet prior, for categorical likelihood which encourages having multi-modal posteriori. paper demonstrates couple of techniques  to apply the reparametrization trick on Dirichlet distribution, by using sum of iid Gamma random variables.

- used method of moments estimator to update the hyper parameter of the Dirichlet distribution which helps to have closer approximation of log likelihood. They update hyper-parameters after every few updates of VAE parameters.

-discussed how to overcome  Stick-breaking VAE “component collapse” issue. Experiments show superior results on supervised and semi supervised, and authors claimed the main reason of this superiority being due to not having disadvantage of component collapse which happens in SBVAE.


Quality and Novelty:
claims in paper are supported by proofs and/or experimental results and there does not exist significant technical issues with the details of claims made in this paper and proofs provided. There are following issues with novelty and quality of paper that I would like discuss them under following three points:

- Authors need to be clear about the motivation of the paper, if the motivation of the paper is to encourage the multi-modality in posteriori distribution, using Gaussian prior and methods like normalizing flow Rezende, Danilo Jimenez, and Shakir Mohamed. "Variational inference with normalizing flows." arXiv preprint arXiv:1505.05770 (2015) or similar may be able to do the same work in which case paper should compare its results to those ideas which has not been done in this paper.

- second appealing point that this paper can make is to use Dirichlet prior for the purposes like community detection, topic modeling and LDA  etc etc. In this case, I did not find significant difference between the proposed method and what is found in Srivastava, Akash, and Charles Sutton. "Autoencoding variational inference for topic models." arXiv preprint arXiv:1703.01488 (2017), but due to the encourages of multi-modality authors show in average DirVAE performs better in measures like perplexity and NPMI. Under this condition, my main concern is interpretablity of posteriori. That will be discussed under next point

- Main motivation behind using Dirichlet prior, is to have posteriori with a few significant related topic and many unrelated topic for every word. By changing the concentration parameter in stick-breaking, it is possible that performance of stick-breaking method increase in perplexity and NPMI scores in cost of loosing interpretability of the model. So having higher concentration parameter can show better performance in the cost of interpretablity that put second point of the paper at risk


Clarity:
The paper is well written and previous relevant methods have been reviewed well. The organization of paper is good, experiments well explained and proofs and mathematical reasoning are clear.



Significance of experiments:
As discussed,in previous sections, the results show superior performance and compared to other methods on semi-supervised and supervised classification on different datasets. Also it has shown in average better perplexity and NPMI score for topic modeling, the only issue can be these scores come as cost of interpretablity of the model. Also it is possible that other competing models can be matching to this results if they do not aim for sparse posteriori.

---

> ### Author Response · Authors · 2018-11-11
> **Responses to Reviewer3.**
>
> Thank you for your review.
>
> Currently, we are doing additional experiments to respond to your constructive comments. Sorry for the delay, but we will give you proper responds to your review with results as soon as possible. Up to the current status of experiments, the VAE with the normalizing flow still suffers from the decoder weight collapsing problem. Hence, the performance would not match to our approach, but we are going to make a certain on this premature result with the experiments, next few days.
>
> Sincerely.

---

> ### Author Response · Authors · 2018-11-16
> **Responses to Reviewer3 with additional experiments.**
>
> To respond to your comments on quality & novelty part, we did additional experiments, and some figures and tables are added or modified.
>
> 1. Regarding the paper "Variational inference with normalizing flows" (Rezende et al.), we add followings: Table 5, Figure 5, and Table 6. In our experimental setting, the performances on pure VAE of GVAE and GVAE-NF20 were barely different, and DirVAE shows discriminative results compared to the both GVAEs. Not only the DirVAE gives better quantitative results as in Table 5 and 6, but it also has better learned latent representations which can be supported by t-SNE visualization such as Figure 3(b) and 5(b), and the decoder weight collapsing of GVAE-NF20 is one reason for such worse quality.
>
> 2. We add SBVAE augmentation on topic models as a baseline in the topic modeling experiments. The related results can be found in Table 4, Figure 7, 8, and 9. As you can see, SBVAE augmentation is better than the original model in some sense, and for a certain case, it is quite comparable. However, still, the DirVAE augmentation shows better results in most of cases in terms of datasets and the performance measures.
>
> 3. To show the interpretability of DirVAE augmentation on topic models, we add Table 8 which lists top probability words per topic. We manually re-ordered and put the topics together if there are similar semantic meanings. Also, Figure 7, 8, and 9 shows that DirVAE augmentation on topic models brought better learned latent representation than others, as in the case of pure VAE experiments.
>
> Thanks again for your comments.
>
> Sincerely.

---

> > ### Comment · AnonReviewer3 · 2018-11-28
> > **Respond to your fixes**
> >
> > Thanks for your comments and sorry for late reply, I needed to reread the whole paper again to see how cohesive are different sections of paper and what specific message is tried to be delivered.
> > 1-  I think something should have gone wrong with your experiments with normalizing flow, the negative marginal log likelihood reported to on  Rezende, Danilo Jimenez, and Shakir Mohamed. "Variational inference with normalizing flows." arXiv preprint arXiv:1505.05770 (2015) is 86.5 on MNIST data or less and it is better than GAVE and makes sense to be better than GVAE, so I really think that you need to reconsider your reported results, then comparisons could be correct from here on
> > 2- I have no argument that you can get related word for a topic the discussion was over sparsity and interpretability. In your model latent  parameter sparsity was not imposed so it will have e.g. many topics for each word if applied directly to the topic modeling problem.
> > 3- I have big concern about the massage that this paper carries. If paper is about to solve the  multimodality of posteriori distribution, the message and experimints should be tailored around that message. If the focus is to present the paper more as an application paper which makes improvement over different area, the message should be put out differently.
> > I really like the extended experiments and improvement which are reported in different settings, but as other reviewer and i are concerned with novelty and I can not see the cohesive message is carried by this paper, I do not feel that any further comment can improve the current paper. Unless with unique message in mind the paper is written and experiments are done around the unique message.
> > again, Although I like the paper based on the extend of experiments, i am more intended to mild reject the paper as there are issue with having a message to be carried, how paper is presented and novelty.
> > Thanks and sorry

---

> > > ### Author Response · Authors · 2018-11-29
> > > **Thank you for your review.**
> > >
> > > Dear Reviewer3:
> > >
> > > We respond to your point in the below.
> > >
> > >
> > > 1:
> > > We checked the experiment with the 'normalizing flows' approach, but we were not able to find an error. It should be noted that the experiment code is our creation, but from the authors of the normalizing flows. With our creation, we also tested OMNIGLOT by comparing the normalizing flow and the Gaussian VAE, and we were able to see that the normalizing flow is significantly better than the Gaussian VAE. Also, in this OMNIGLOT experiment, DirVAE was better than the normalizing flow. I think that MNIST is too simple dataset to judge the validation of our implementation of the normalizing flow.
> > >
> > > For a fair comparison, I think that the implementations of both cases should not include techniques, such as Max-out. When we remove these techniques besides the core of the models, it is natural to see a different result compared to the previous literature. I think that these conflict or practices of conducting experiments should be discussed within our research community.
> > >
> > >
> > > 2:
> > > I think that we had a miscommunication between us and Reviewer3. We interpreted the "interpretability" as a further qualitative analysis on the topic model results, which can be discovered in the previous literatures. Often, a topic model is evaluated by a qualitative observation on the topic words and a quantitative measure of the held-out log-likelihood or perplexity. We think that our paper delivers these evaluation aspects.
> > >
> > > Moreover, our paper introduces how to put a prior over a latent variable of a variational autoencoder. This prior will determine the characteristics of the latent variable to a certain extent. The prior of Dirichlet distribution, which we choose, is a typical prior to handling a sparse dataset, so we have not been suspecting that Reviewer3 was questioning such characteristics.
> > >
> > > Evaluating the model fitness to a dataset generated from a sparse latent variable can be answered by calculating the quantitative evaluation point, such as the held-out perplexity or the log-likelihood. If a model cannot capture the inherent sparsity, the quantitative measurement will be bad. From this perspective, DirVAE improves the quantitative aspect of the neural topic models in general.
> > >
> > >
> > > 3:
> > > In some part, I agree with you that this paper can be regarded as a "distracted" paper. However, from our perspective, we have had a consistent motivation of this paper.
> > >
> > > Our motivation is "An explicit deep generative model should further extend the choices of the probability distributions and their causal relation modelings to reflect the true latent dynamics of the generative process on the modeled domain." For example, many prior works, such as LDA, have succeeded because of the careful choices of probability distributions and causal relations to capture the text generation process. It should be noted that the only difference between LDA and pLSI is the Bayesian treatment with a Dirichlet prior. Then, it is natural that the explicit DGM community needs to extend and improve the modeling of the prior and the latent variables by accepting distributions other than the Gaussian. Our paper provides a methodology of modeling the Dirichlet prior in the VAE setting and an improvement of its usage in the text domain. From these logical flow, we consider the message is fairly simple.
> > >
> > > Having said that, the discussion on the component collapsing and the evaluation on the classifiers could be distracting. On the other hand, the enabled priors are always investigated in its properties, so we had to include those discussions.
> > >
> > >
> > > Again, we appreciate your reviewer service to our research community.

---

> ### Author Response · Authors · 2018-11-24
> **t-SNE figures are modified.**
>
> For better representation, we modified all t-SNE figures in the paper.
>
> Sincerely.

---

### Meta-Review · Area_Chair1 · 2018-12-12
**Dirichlet Variational Autoencoder**

**Confidence:** 4
**Recommendation:** Reject

**Metareview:**

This paper applies Dirichlet distribution to the latent variables of a VAE in order to address the component collapsing issues for categorical probabilities. The method is clearly presented, and extensive experiments are carried out to prove the advantage against VAEs with other prior distributions.

The main concern of the paper is the limited novelty. The main methodology contribution of this paper is to combine the decomposition a Dirichlet distribution as Gamma distributions, and approximating Gamma component with inverse Gamma CDF, but both components are common practices.

R3 also points out that the paper is distracted by two different messages the authors try to convey. The presentation and experiments are not designed to provide a cohesive message. The concern is not solved in the authors' feedback.

Based on the current reviews, this paper does not meet the standard for ICLR publication. Despite the limited novelty in the proposed model, if the paper could be revised to show that a simple modification is good for solve one problem with general applications, it would make a good publication in a future venue.